# Clinical Cognitive Motor Dissociation: A Case Report Showing How Pitfalls Can Hinder Early Clinical Detection of Awareness

**DOI:** 10.3390/brainsci12020157

**Published:** 2022-01-25

**Authors:** Jane Jöhr, Viviana Aureli, Ivo Meyer, Giulia Cossu, Karin Diserens

**Affiliations:** 1Acute Neuro-Rehabilitation Unit, Service of Neurology, Department of Clinical Neurosciences, Lausanne University Hospital (CHUV) and University of Lausanne (UNIL), 1011 Lausanne, Switzerland; ivo.meyer@chuv.ch (I.M.); karin.diserens@chuv.ch (K.D.); 2Service of Neurosurgery, Department of Clinical Neurosciences, Lausanne University Hospital (CHUV) and University of Lausanne (UNIL), 1011 Lausanne, Switzerland; viviana.aureli@chuv.ch (V.A.); giulia.cossu@chuv.ch (G.C.)

**Keywords:** brain injury, cognitive motor dissociation, disorders of consciousness, motor behaviour tool, clinical diagnosis, case report

## Abstract

This study presents the case of a brain-injured patient whose pathological awakening after coma and absence of interaction led to a diagnosis of lack of consciousness when standard clinical scales were administered. However, we were able to demonstrate conscious perception in this patient from initial clinical assessments using the Motor Behaviour Tool in the acute stage, complemented by a systematic search for potential obstacles blocking his execution of motor responses (pitfalls). This refinement of the diagnosis enabled prediction of a favourable outcome despite the severity of the lesions, with the patient’s evolution confirming our prediction. Faced with an unresponsive patient, every specialist should go beyond the absence of response with the standard scores, consider the possibility of a hidden consciousness and look for rigorous ways of proving it.

## 1. Background

Over the last 15 years, specialized neurobehavioural rating scales have been developed to provide a reliable means of assessing consciousness. The Coma Recovery Scale-Revised (CRS-R) is currently used as the gold standard in the diagnosis of disorders of consciousness (DOC) in the subacute and chronic settings [1]. The downside of this scale is that it requires consistent motor output and therefore may be insufficient for assessing a cognitive ability to interact in patients with impaired motor efference/output, a condition described as cognitive motor dissociation (CMD) [2]. Indeed, the literature shows that the rate of misdiagnosis of conscious patients as unconscious remains high (30–40%) and such errors may present serious consequences as true severe altered consciousness carries unfavourable prognosis [3].

It has been shown by our group that the Motor Behaviour Tool-revised (MBTr) can reveal residual cognition in almost two-thirds of acute patients diagnosed as behaviourally unresponsive by the CRS-R [4]. The MBTr tool was conceived to unmask clinically a conscious perception (i.e., defined as clinical CMD) by identification of subtle motor behaviour undermined by the CRS-R, thereby allowing a more accurate assessment of consciousness and determining a potential for better recovery [5]. The MBTr is divided into two parts, which test nine clinical signs: seven positive signs hinting at intact awareness (from five categories) and two negative signs, representing ‘red flags’, reflecting strategic pyramidal tract and/or brainstem lesions or unveiling a true awareness impairment due to widespread cortico-thalamic lesions. On a pathophysiological level CMD reflects strategic lesions impairing the motor output channels. Patients with CMD can be diagnosed clinically because the likelihood of a complete disruption of all motor output channels is low. In fact, if many motor output channels are disrupted, the patient is probably suffering from widespread lesions, causing not only true impairments of awareness (no positive MBTr signs) but tell-tale signs of DOC (negative MBTr signs), such as stereotyped posturing (in the absence of strategic corticospinal lesions on imaging). Therefore, subtle signs of intentionality after adequate stimulation can be detected by means of the MBTr in most clinical CMD patients. Furthermore, the sensitivity of this tool is unaffected by the presence of confounding clinical factors (i.e., pitfalls) that interfere with sensory and/or motor afference or intrinsic brain activity [6].

Here we describe the case of a brain-injured patient presenting with several pitfalls and consequently diagnosed as unresponsive based on the CRS-R only, while clear signs of conscious perception were identified by means of the MBTr.

## 2. Case Presentation

### 2.1. History

A 53-year-old patient with no previous medical or psychiatric history was admitted to our emergency department after a severe brain injury (Glasgow Coma Scale 3/15) following a car accident. The patient needed endotracheal intubation and sedation, and his initial clinical examination showed reactive symmetrical pupils. An emergency computed tomography (CT) total body scan revealed numerous lesions: inferior vena cava laceration, abdominal bleeding and left inferior limb fractures. A cerebral CT-scan revealed a right parietal sub-arachnoid haemorrhage and left maxillary bone fracture. There were no signs of brain swelling and basal cisterns were free. As an emergency surgery was required to treat the inferior limb fracture and abdominal bleeding, an intracranial pressure monitoring was implanted to drive the neuro-reanimation. An intensive neuro-reanimation was required over the following days to attempt to get values of intracranial pressure below 15 mmHg. After 48 h, despite a maximal neuro-reanimation treatment with curare and hypertonic solution, intracranial pressure was refractory to medical treatment and a surgical treatment through right decompressive craniectomy was decided, which resulted in the normalization of intracranial pressure. Strict neuro-reanimation procedures continued for 3 days, and sedation was withdrawn 10 days post injury. A tracheotomy was performed 10 days later.

### 2.2. Clinical Examination

Twenty-four hours post-sedation withdrawal, the patient presented a pathological awakening with absent external responsiveness to stimulation. Hence, he was clinically assessed by two experienced physicians by means of the CRS-R and the MBTr (see details in Table 1). On the first two assessments, the patient was classified as being in a state of coma. Then, the opening of the eyelids, oral reflexes and a startle reflex to noise classified him as unresponsive wakefulness syndrome (UWS) according to the CRS-R, which corresponds to a state of unconsciousness. It was not until the eighth assessment (44 days after onset) that the CRS-R detected a minimal sign of consciousness (i.e., visual pursuit) and categorized him as being in a minimally conscious state minus (MCS-). However, the MBTr identified him as presenting with clinical cognitive motor dissociation as early as the fifth evaluation (24 days after onset), with two positive signs of conscious perception; i.e., an intentional limb retraction considered as a defence gesture on painful stimulation of the extremities and an associated grimace with slight movement of the head towards the stimulation. At least one positive sign is required to categorize a patient with clinical CMD (see Appendix A for a complete list of MBTr items). The withdrawal response to pain was considered intentional both in character and for the muscles involved. To be more specific: the kinematics of an intentional defence gesture were defined in opposition to those of the nociceptive withdrawal reflex. In the upper limb, a nociceptive stimulation well above the pain threshold applied to the index finger elicits a reflex movement consisting of wrist adduction (frontal plane), elbow flexion (sagittal plane), and shoulder anteflexion (sagittal plane). A response clearly differing from these reflex movements was observed by two well-trained examiners. In addition, the MBTr considers facial grimacing after painful stimulation as a positive sign of interaction, even in the absence of a motor response. As originally described, an appropriate response may be accompanied by a facial grimace or a generalized increase in movement [7]. Furthermore, the observations of subtle behaviour considered intentional were validated by paraclinical examinations revealing the presence of interaction-limiting pitfalls (see the following section), in a patient whose remaining imagery did not support a non-responsive state. Neurological examination showed tetraparesis, the absence of deep tendon reflexes on both sides and the absence of plantar reflex on the right side. Moreover, an abnormal oculomotor pattern (horizontal nystagmus-type movements, irregular and dysconjugate) was noted.

### 2.3. Pitfall Identification Process

We adopted a strategy of searching for potential pitfalls concomitant with the neuro-behavioural assessment using an adapted flowchart (Figure 1). These pitfalls are either the main reason for the lack of interaction (e.g., akinetic mutism) or confounding factors additionally masking residual awareness (e.g., cortical blindness, NCSE). In most cases, a combination of factors contributes to the seemingly non-responsiveness of the patient. We have previously demonstrated that there are imaging patterns showing circumscribed but strategic damage to the motor efferent system with preserved cortico-subcortical connectivity in clinical CMD as opposed to widespread cortico-thalamic damage typical of true DOC [8]. The flowchart compares the clinical examination with the results of the structural imaging and neurophysiological examinations to search for neurological deficits intrinsically related to the brain injury process and interfering with the production of appropriate behavioural or motor responses. This allows us to establish a precise diagnosis of the presence of conscious perception in a patient who does not interact according to the standard scales. In this case, we suspected a thalamic impairment explaining the attentional disorders, a frontal akinetic syndrome and the presence of a polyneuropathy justifying the global akinesia as pitfalls masking the conscious perception.

A neuroradiological evaluation confirmed these suspicions. MRI of the brain revealed a discrete extra-cranial cerebral herniation, perivenular haemorrhages of the left frontal cortical-subcortical junction and the corpus callosum suggesting traumatic haemorrhagic axonal lesions. There was also a restriction in the diffusion images of the splenium of the corpus callosum and the right cerebral peduncle, suggesting non-haemorrhagic traumatic axonal lesions. Finally, small oedemato-haemorrhagic contusion in the left orbito-frontal, left fusiform gyrus and right cerebellum were reported (Figure 2).

^18^F-FDG PET revealed significant diffuse cortical hypometabolism including the primary sensorimotor areas, with less damage to bilateral insular regions, associated with bilateral hypometabolism of subcortical structures (striatum and thalamus), alongside diffuse hypometabolism of the cerebellum.

Electroneuromyography (ENMG) demonstrated a reduction in amplitude of the motor responses by stimulation of the peroneal and musculocutaneous nerves. The moderate abnormalities observed were compatible with a mild critical illness polyneuropathy yet did not account entirely for the tetraplegia observed in the patient.

### 2.4. Therapeutic Interventions, Outcomes at Discharge and Follow-Up

Over the 6-week period in the acute neuro-rehabilitation unit, the patient underwent an individualized intensive multidisciplinary rehabilitation program.

The neurological examination and outcome at discharge are detailed in Table 1. At the neuropsychological level, the patient had a dysexecutive syndrome encompassing behavioural and speech aspontaneity, associated with severe attentional impairments. Although hypophonic, he was able to communicate on a quasi-functional level by expressing his needs with short statements and gestures and by consistently answering simple closed-ended questions. He understood simple commands but required sustained stimulation when complexity increased. At the functional level, the patient was partially independent and could safely perform self-care activities (e.g., shaving, washing upper body and applying deodorant and perfume). His rehabilitation potential was deemed good at discharge, and he was referred to a neurorehabilitation centre. Length of stay in rehabilitation unit was 43 days with a Glasgow Outcome Scale score at discharge of 3 indicating severe disability. The tracheotomy was weaned 58 days post-injury and swallowing disorders had improved. An autologous bone flap cranioplasty was performed 51 days post-craniectomy without post-operative complications. A left hip prosthesis was placed three-and-a-half months post discharge, and he was able to walk again with the aid of a rollator.

## 3. Discussion

This case study presents the history of a patient whose pathological awakening after coma and lack of interaction led to a diagnosis of absence of consciousness when only the standard scales were administered. Nonetheless, we were able to demonstrate conscious perception in this patient from early clinical assessments in the acute stage using our specialized motor behaviour observation tool, the MBTr, complemented by a systematic search for pitfalls that could hinder the production of motor responses. The refinement of the diagnosis helped to predict a favourable outcome despite the severity of the lesions, and the patient’s evolution confirmed our prediction.

Our case illustrates several important points in the pursuit of an accurate and correct diagnosis of consciousness disorders in the acute phase. Firstly, the presence of conscious perception is correlated with a better prognosis, which contributes directly to decisions made in the intensive care unit about whether or not to continue care and the information given to the families [9].

Secondly, the identification of the pitfalls hindering the adequacy of motor responses that may then totally or partially mask a conscious perception in a patient appearing unconscious is essential. Indeed, in a retrospective cohort study we previously demonstrated a high prevalence of patients presenting confounding factors leading to misdiagnosis of their state of consciousness [6].

In the case described here, we found three major coexisting pitfalls, which deserve further consideration. The prominent deafferentation of the subcortical structures (thalamus and striatum) are associated with severe damage to the regulation of arousal and functional integration of the forebrain, explaining the significant fluctuations in the patient’s alertness [10]. In addition, dysfunction of the thalamocortical network involved in motor control may also explain the lack of external responsiveness in this patient based solely on the standardized scales [11].

The frontal syndrome coming from disconnections in the frontal cortico-subcortical motor pathways determines a condition close to akinetic mutism accounting for the absence of behavioural reactions despite sustained stimulation [12]. Moreover, white matter lesions including the wide range of projections from the supplementary motor area to the striatum as well as its direct connections to Broca’s area, which may further provide the anatomical basis explaining akinesia and reduced speech [13].

These interpretations may be also based on the mesocircuit model according to which CMD patients present a functional impairment of the forebrain systems associated with motor preparation and action [14], differentiating them from true DOC where extensive cortico-thalamic damage is typical [15].

Finally, previous studies showed that 40–80% of critically ill people presented with acute polyneuromyopathy [16], motivating a careful evaluation of peripheral nerve and muscle function.

The present case study has several limitations. First, we categorized the patient as presenting with clinical CMD only on a clinical rating from MBTr evaluation. We did not perform an active mental-imagery task to confirm the clinical diagnosis (original operational definition of CMD [2]). Further studies combining MBTr and functional MRI or EEG testing are required to provide an objective measure of intact awareness. Second, we identified the presence of presumed pitfalls at a very early stage using clinical/paraclinical criteria. However, these criteria are routinely used in our institution, and the follow-up observation of the patient’s functional and neuropsychological outcome confirmed the initial identification. Finally, we describe here a single patient case, which limits the applicability of the results to similar cases. In this perspective, this report can be considered as a pilot study for larger-scale research.

## 4. Conclusions

This case illustrates the importance of establishing a detailed and structured diagnostic procedure in early clinical detection of consciousness using a simple clinical observation tool combined with a comprehensive neurological examination of the primary neurological pathways complemented by a systematic search for pitfalls hindering responses. Faced with an unresponsive patient, and despite the necessary rapidity of clinical examinations in the acute stage, each specialist should imperatively weigh up their diagnosis, consider options other than the standard scales and base their evaluation on all the elements that are possible to acquire. The absence of proof is not proof of absence, and a wrong diagnosis may lead to ethical and prognostic implications of great importance.

## Figures and Tables

**Figure 1 brainsci-12-00157-f001:**
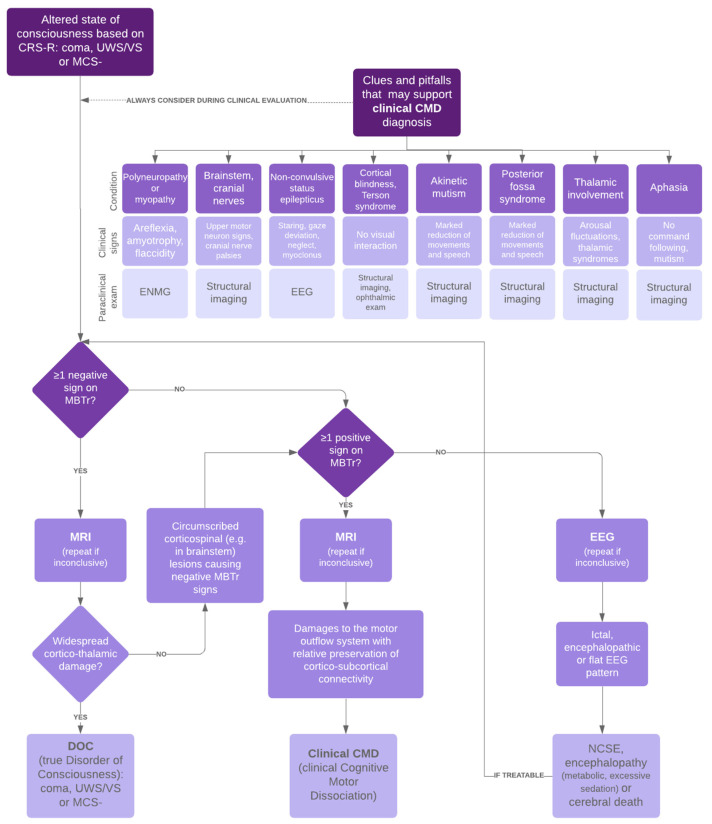
Acute assessment of patients with major cerebral impairment. CRS-R Coma Recovery Scale-Revised, UWS/VS unresponsive wakefulness syndrome/vegetative state, MCS- minimally conscious state minus, clinical CMD clinical cognitive motor dissociation, ENMG electromyoneurography, EEG electroencephalography, MBTr Motor Behaviour Tool-revised, MRI magnetic resonance imaging, DOC disorders of consciousness, NCSE non-convulsive status epilepticus.

**Figure 2 brainsci-12-00157-f002:**
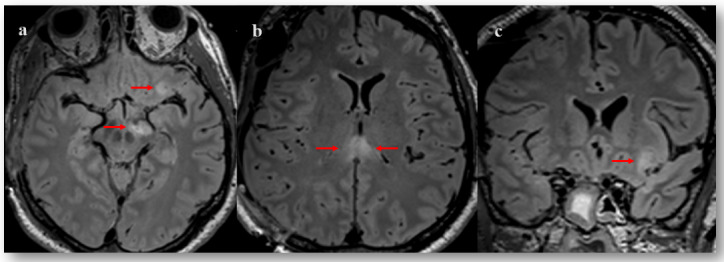
Figure 2. Fluid attenuated inversion recovery (FLAIR) magnetic resonance imaging taken 13 days post-severe TBI. Axial and coronal sections showing areas of hyperintense signal (indicated with arrows) in the left cerebral peduncle (**a**), orbito-frontal cortex (**a**,**c**) and splenium of corpus callosum (**b**).

**Table 1 brainsci-12-00157-t001:** Clinical features and outcomes.

**Clinical Assessment Post-Sedation Withdrawal to Discharge from Acute Rehabilitation Unit**			
**Time after Onset (days)**	**CRS-R Diagnosis**	**CRS-R Subscale Scores**	**MBTr Classification**
9	Coma	A0-V0-M0-O0-C0-Ar0	
11	Coma	A0-V0-M0-O0-C0-Ar0	
16	UWS	A0-V0-M0-O1-C0-Ar1	
23	UWS	A1-V0-M0-O1-C0-Ar1	
24	UWS	A1-V0-M2-O1-C0-Ar1	Clinical CMD with 2 positive signs
44	MCS-	A2-V3-M2-O2-C0-Ar2	
51	MCS+	A3-V4-M5-O2-C1-Ar2	
58	MCS+	A3-V4-M5-O3-C1-Ar2	
65	EMCS	A4-V5-M6-O3-C2-Ar2	
**Neurological examination and outcome at discharge**			
**Neurological exam**			
**Mental status**		**Cranial nerve Sensory system**	**Motor/Functional level**
Dysexecutive syndrome with loss of spontaneity		Oculomotricity preserved in all planes	Upper-right limb functional in distal part
Speech and comprehension disorders		Absence of paresis or facial hypoesthesia	Upper-left limb improved in strength
Attentional disturbances		Sensation coarsely preserved in 4 limbs	Right-lower limb functional
Disorientation in space			Mobilization of left-lower limb limited

CRS-R Coma Recovery Scale-Revised, UWS unresponsive wakefulness syndrome, MCS- minimally conscious state minus, MCS+ minimally conscious state plus, EMCS emergence from minimally conscious state, clinical CMD clinical cognitive motor dissociation. The subscales for the CRS-R are Auditory Function (A), Visual Function (V), Motor Function (M), Oromotor Function (O), Communication (C), and Arousal (Ar).

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
