# Peer review of "Clinical Cognitive Motor Dissociation: A Case Report Showing How Pitfalls Can Hinder Early Clinical Detection of Awareness"

_brainsci, 2022, doi:10.3390/brainsci12020157_

Round 1

Reviewer 1 Report

Review of

Jöhr et al., "Clinical Cognitive-Motor Dissociation..."

Because Locked-in-Syndrome generally, and Cognitive-Motor Dissociation particularly, is a rare condition, even single case reports are useful. The current report is particularly interesting as it presents an innovative approach potentially important for a number of patients with severe neurological abnormalities. Nevertheless, I would suggest some necessary improvements.

To begin with, the abbreviation cCMD is plainly incorrect. First of all, we should not enforce doctors to use special signs when describing their patients; rather, normal English (German, French etcetera) alphabet should suffice. Second, medical diagnoses are not determined by assessment tools, otherwise we will soon get ctCMD, mrtCMD, eegCMD, p300CMD, megCMD, fnirsCMD and many other awkward things. Third and most important, why should we talk about specifically clinical CMD, when we clearly see from both Figure 1 and the main text that several non-clinical techniques substantially contribute to the diagnosis. As a scientific/medical terminus, CMD is good enough.

The authors put the validity of CRS-r in question, and this is OK. However, if there is no golden standard in the diagnostics, the most important criterion is prognosis. Therefore, much more than a few lines in the bottom of Table 1 should be said about the state of the patient at discharge. What was attained during rehabilitation? Was reliable communication with the patient possible? Was he able to use some simple tools, e.g. to camb himself?

Finally, a most general question remains open as how the authors define the concept of CMD. Usually, it is described as the preserved cognition without the ability to follow motor commands a required in most neurological tests such as CRS-r. As far as I understood the MS, and specifically Table 1, the depicted patient had severe cognitive impairment down to the level of simple sensation. This should be clarified. What level on the scale of cognitive imparment should be sufficient to still set the diagnosis of CMD ("preserved cognition")? Particularly, what are the proofs that the two observed positive symptoms ("limb retraction considered as a defence gesture on painful stimulation ... and an associated grimace with slight movement of the head towards the stimulation") were cognitively mediated intentional movements and not simple reflexes?

Reviewer 2 Report

The case study is interesting, well studied and well written.

However, I have some doubts on a couple of passages.

In the introduction (page 1) it would be important to know the items of the Motor Behaviour Tool-revised (MBTr), as the manuscript is focused on the usefulness of this tool.

The flowchart reported in figure 1 looks very important but it completely lacks explanation.

For instance, ‘non-convulsive status epilepticus’: usually seizures are not constant and do not represent a bias for Cognitive Motor Dissociation (CMD); it is not clear why seizures are associated to neglect.

Another point to explain is cortical blindness: it is typically observed in patients who can speak.

Regarding the second part of the flowchart, potentially it represents an important reasoning for the clinician assessing a patient in the acute phase. However, again, it is necessary that the flowchart is explained in the text.

In the first column, also a brainstem lesion can cause DOC, and it is not clear the meaning of ‘strategic’. In addition, from the diagram it seems that if a patient has positive signs he/she cannot be in VS.

It would be important to clarify the thalamo-cortico-thalamic connectivity. It is also possible that it cannot be observed/detected with MRI in the acute phase.

Minor point

Table 1: as it is reported, it is not clear when CMD was given to the patient

Round 2

Reviewer 1 Report

To point 1: I do not see the necessity to introduce as many new terms as possible. For a practical neurologist it is not a help but rather disorientation. Fully unjustified is inventing a special sign like  ͨ or ˁ instead of a simple letter "c".

To point 2: The authors' response is adequate.

To point 3: The authors justify their conclusion, that the observed responses were cognitively mediated, by neuromorphological, theoretical neurological, prognostic, philosophical, and clinical arguments. Unfortunately, the reasons are presented in the answer letter in a somewhat confused form. I suggest that the authors order the arguments correctly and present them in a concise form in the manuscript itself, not only in the letter. From my viewpoint, clinical arguments are the most important.
